# Calreticulin—Enigmatic Discovery

**DOI:** 10.3390/biom14070866

**Published:** 2024-07-19

**Authors:** Gillian C. Okura, Alamelu G. Bharadwaj, David M. Waisman

**Affiliations:** 1Department of Pathology, Dalhousie University, Halifax, NS B3H 1X5, Canada; gillian.okura@dal.ca (G.C.O.); alamelu.bharadwaj@dal.ca (A.G.B.); 2Department of Biochemistry and Molecular Biology, Dalhousie University, Halifax, NS B3H 1X5, Canada

**Keywords:** calreticulin, calregulin, high-affinity calcium-binding protein (HACBP), calsequestrin, essential thrombocythemia, sarcoplasmic reticulum, endoplasmic reticulum, Michalak

## Abstract

Calreticulin (CRT) is an intrinsically disordered multifunctional protein that plays essential roles intra-and extra-cellularly. The Michalak laboratory has proposed that CRT was initially identified in 1974 by the MacLennan laboratory as the high-affinity Ca^2+^-binding protein (HACBP) of the sarcoplasmic reticulin (SR). This widely accepted belief has been ingrained in the scientific literature but has never been rigorously tested. In our report, we have undertaken a comprehensive reexamination of this assumption by meticulously examining the majority of published studies that present a proteomic analysis of the SR. These analyses have utilized proteomic analysis of purified SR preparations or purified components of the SR, namely the longitudinal tubules and junctional terminal cisternae. These studies have consistently failed to detect the HACBP or CRT in skeletal muscle SR. We propose that the existence of the HACBP has failed the test of reproducibility and should be retired to the annals of antiquity. Therefore, the scientific dogma that the HACBP and CRT are identical proteins is a non sequitur.

## 1. Identification of the HACBP

The MacLennan laboratory was responsible for many key discoveries concerning the biochemistry of muscle contraction, particularly sarcoplasmic reticulum (SR) function [1]. Dr. MacLennan’s discovery of the Ca^2+^-ATPase pump that controls calcium sequestration by the SR was a fundamental discovery that provided the key to understanding the mechanism of muscle relaxation. His lab systematically characterized the components of the SR, including the ryanodine receptor, which acts as a Ca^2+^ release channel to activate muscle contraction, and calsequestrin, which acts as a significant Ca^2+^ buffer in the SR [2].

In their pivotal 1971 publication [3], MacLennan’s group analyzed the protein components of the rabbit skeletal muscle SR and reported the presence of a Ca^2+^-binding protein of molecular weight 44 kDa by sodium dodecyl sulfate-polyacrylamide gel electrophoresis (SDS-PAGE), which was purified and shown to be 42 kDa by gel permeation chromatography [3]. This protein was named calsequestrin, and subsequent studies have identified it as a fundamental regulator of Ca^2+^ homeostasis in the skeletal muscle SR.

In 1973, MacLennan’s group reported a detailed analysis of the major proteins of rabbit skeletal muscle SR. They observed by SDS-PAGE analysis that the solubilized protein fraction from purified rabbit skeletal muscle SR was comprised of five Coomassie blue-staining acidic proteins of 102 kDa (ATPase), 54 kDa, and 44 kDa (calsequestrin) and three acidic proteins between 32 and 20 kDa [4]. Calsequestrin was identified as a dominant protein. Next, these investigators used lactoperoxidase to label the SR fraction with ^125^I, performed SDS-PAGE analysis, and reported the relative quantity of each protein [4]. These values were reported as ATPase (1536 cpm), 54 kDa (156 cpm), calsequestrin (254 cpm), and acidic proteins (1409 cpm). This suggested that the 54 kDa protein was present in the SR at more than 50% of the calsequestrin levels. The 54 kDa protein was purified by several chromatography steps, and the amino acid composition was documented. These investigators also reported that the 54 kDa protein bound Ca^2+^ with an approximate Kd of 100 μM.

In 1974, MacLennan’s group analyzed the protein components of this highly purified rabbit SR preparation in more detail [5]. They reported that analysis by SDS-PAGE, according to the method of Weber and Osborn [6], revealed the presence of five major Coomassie blue-staining protein bands (Figure 1). These bands were identified as the ATPase, M_r_ = 100,000; the High-Affinity Ca^2+^-Binding Protein (HACBP), M_r_ = 55,000; calsequestrin, M_r_ = 44,000; three acidic proteins of M_r_ 32–20,000; and a proteolipid, M_r_ = 12,000 (Figure 1).

The MacLennan lab purified the 54 kDa protein by successive chromatography steps and pooling the appropriate chromatographic fractions according to molecular weight (55 kDa). They considered the purified protein purely based on SDS-PAGE analysis performed according to Weber and Osborn and characterized this protein in more detail. They named a pooled protein peak from column chromatography the HACBP and revised its molecular weight to 55 kDa by SDS-PAGE [5]. They reported the yield of the purified protein as 1 mg/800 g skeletal muscle tissue. Although the recovery after purification was not reported, it would be expected that purification by four chromatography steps would result in a 10–20% recovery, suggesting 6–12 mg HACBP/kg. This indicates that the HACBP has a reasonable abundance in the SR. They also reexamined the Ca^2+^-binding properties of the HACBP and reported that in the presence of KCl, the HACBP bound one mol Ca^2+^ with a K_d_ of 2.5–4 μM. Later, they noted that the HACBP was responsible for 5–10% of the total protein of the rabbit skeletal muscle SR [7].

This report contradicted the publication by Ikemoto, who used SDS-PAGE to identify Coomassie blue-staining proteins of 125–165 kDa, 100 kDa (ATPase), and 62 kDa as the major proteins of the SR [8,9]. In subsequent publications, the 62 kDa protein was called the calcium precipitable protein, Cp. Ikemoto reported a more detailed analysis of the SR protein and determined the CP molecular weight to be 55 kDa [8,9]. Ikemoto procured a sample of calsequestrin from MacLennan’s group and reported that, in contrast to MacLennan’s report of 44 kDa for calsequestrin, the CP and calsequestrin moved with identical mobility in SDS-PAGE, i.e., 55 kDa [9]. Ikomoto’s group detected no Coomassie blue-staining SR proteins corresponding to MacLennan’s 54 kDa protein.

Interestingly, MacLennan obtained a sample of the 55 kDa CP protein from Ikemoto’s laboratory and reported that the 55 kDa protein and the 44 kDa protein from his laboratory both migrated in his system with a molecular weight of 44 kDa [5]. Therefore, both laboratories concluded that both proteins were calsequestrin. The issue of why Ikemoto’s laboratory did not detect a Coomassie blue-staining protein band migrating between calsequestrin and the 100,000-dalton ATPase, the position on the gel corresponding to the MacLennan’s 54 kDa protein, was not discussed by these researchers.

In 1980, Michalak, Campbell, and MacLennan further detailed their analysis of the proteins of the rabbit skeletal muscle SR [10]. They reported that the molecular weight of the 44 kDa protein calsequestrin varied according to the SDS-PAGE procedure. When analyzed by the method of Weber and Osborn [6], the Coomassie blue-staining intensity ratio of the protein bands at 44 kDa containing calsequestrin and at 55 kDa containing the HACBP was about 2/1. In SDS-PAGE, run in the second dimension according to the method of Laemmli [11], calsequestrin moved with a lower mobility corresponding to a molecular weight of 63 kDa. The relative staining density between the band containing calsequestrin and the HACBP band changed to about 5/1. This was reported due to the separation of 55 kDa contaminants from the 55 kDa Coomassie blue-staining protein band. They also reported that the ratio of ATPase to HACBP in the purified SR preparation was about 9:1, but in the terminal cisternae SR vesicles (TC), this increased to 6:1 [10]. These measurements of the abundance of the HACBP are important because they indicate that the HACBP was reasonably abundant in the SR. They also produced an antibody to the HACBP and established that the HACBP and calsequestrin were distinct proteins.

In 1983, Kevin Campbell of the MacLennan laboratory also utilized the technique of two-dimensional SDS-PAGE, utilizing Weber and Osborn SDS-PAGE in one dimension and the Laemmli SDS-PAGE in the second dimension, and repeated the analysis that had been performed by Michalak three years earlier. The SR preparation method used by Campbell was the same as the SR preparation used by the MacLennan laboratory in the 1974 publication (Figure 1). Interestingly, as shown in Figure 2, Campbell could not detect the HACBP by SDS-PAGE, nor was the HACBP mentioned in this publication. Specifically, both Ostwald and MacLennan (Figure 1, lane A) and Campbell et al. (Figure 2, lane 3) present similar SDS-PAGE analyses and show a 53–55 kDa Coomassie blue-staining band above the calsequestrin Coomassie blue-staining band. This is the band that Ostwald and MacLennan called the HACBP band. However, when Campbell utilized two-dimensional SDS-PAGE, initially devised by Michalak, to further resolve the 53–55 kDa band, the HACBP was no longer detected. Instead, a 53 kDa Coomassie blue-staining band was observed. This 53 kDa Coomassie blue-staining band was called the 53 kDa glycoprotein. The MacLennan and other laboratories have demonstrated that the 53 kDa protein is distinct from the HACBP. The analysis by Campbell et al. (Figure 2) established that the Ostwald and MacLennan Coomassie blue-staining 55 kDa band called the HACBP (Figure 1) was the 53 kDa glycoprotein.

In 1989, Koch [13] reported the sequence of a 55 kDa protein they initially identified in 1988 as an ER protein called CRP55 [14]. They proposed the name calreticulin (CRT) for this protein. Later, a report from Michalak’s laboratory detailed the complete amino acid sequence of a protein called the HACBP of skeletal muscle SR [15]. This sequence corresponded to the amino-terminal sequence of a hepatic Ca^2+^-binding protein that the Waisman laboratory had purified and called calregulin [16]. Calregulin was initially identified and purified in 1984 by the Waisman laboratory [17]. In 1985, the Waisman laboratory extensively characterized this protein, and immunofluorescence staining demonstrated that this protein localized to the ER protein [18]. The Waisman laboratory further detailed Calregulin in subsequent publications [16,19,20,21].

The Michalak laboratory has claimed that the MacLennan laboratory initially discovered CRT as the HACBP of rabbit skeletal muscle SR in 1974. The analysis presented in 1983 by Campbell (Figure 2) would dispute the validity of Michalak’s claim. In this review, we directly address the discrepancy between the data presented by Campbell and Michalak. In the current report, we show that based on an extensive literature analysis, multiple laboratories, including the MacLennan laboratory, have not reproduced the original 1974 report of the identification and purification of the HACBP in skeletal muscle SR. Therefore, the HACBP has failed the test of reproducibility and is an artefact. We also propose that Michalak did not sequence the HACBP in 1989, as claimed [15], but sequenced CRT, a protein we partially sequenced in 1987 [16]. This review presents the data from other laboratories that show that the HACBP is an artifact and that CRT is not present in skeletal muscle SR.

Furthermore, as discussed in Section 4, the antibody used by Michalak to clone CRT from an expression library was not the same antibody prepared against the HACBP reported in 1974 but was purified against CRT protein isolated from skeletal muscle homogenates. The skeletal muscle homogenates would comprise muscle cells and cells containing CRT, such as fibroblasts and endothelial cells.

## 2. A Reexamination of the Proteins of Skeletal Muscle SR

Multiple laboratories, using various detection techniques ranging from high-resolution SDS-PAGE to mass spectrometric analysis, have been used to identify and characterize the proteins of skeletal muscle SR. These studies have failed to identify the HACBP/CRT in skeletal muscle SR preparations. For example, a high-resolution mass spectrometric analysis of highly purified skeletal muscle SR proteins was published in 2012 [22]. These investigators identified 31 distinct proteins in the SR, and surprisingly, the HACBP or CRT was not detected.

To rigorously resolve whether or not the HACBP exists and the possible relationship between the HACBP and CRT, we have analyzed the results from multiple laboratories that characterized the protein components of the skeletal muscle SR. These publications have utilized Ca^2+^ binding, Zn^2+^ binding, and SDS-PAGE. followed by staining with the dye Stains-All to document the proteins of the SR. Furthermore, since 1972, the sensitivity and resolution of SDS-PAGE have improved, and Coomassie blue staining has been optimized for maximum resolution. If CRT is present in the skeletal muscle SR, then it should be detectable by any of these techniques. Here, we develop the theme that after the 1974 identification of the Coomassie blue-staining 55 kDa band in SR preparations, named the HACBP, only a 1980 publication from the Michalak laboratory has detected the HACBP in the analysis of the proteins of the skeletal muscle SR.

Notably, several laboratories previously identified a Coomassie blue-staining band at 55 kDa in SR preparation. For example, the SR preparation of Fleischer resolved a Coomassie blue-staining 55 kDa band called M55 that was barely detectable [23], constituted about 30% of calsequestrin staining [24], or was about the same in intensity as calsequestrin [25]. Meissner determined the 55 kDa Coomassie blue-staining band to be about 20% of calsequestrin [26], and during the initial solubilization of the SR, calsequestrin and M55 were of similar staining intensity [24]. Interestingly, the relative amount of this band relative to calsequestrin varies considerably among these publications. Other laboratories found calsequestrin to be the 55 kDa Coomassie blue-staining band [8]. As discussed, the analysis of skeletal muscle SR preparations performed in recent years fails to identify the presence of HACBP or CRT in the skeletal muscle SR. However, regardless of the variability in the literature observing the 55 kDa band, the analysis performed by Campbell et al. (Figure 2) clearly shows that this 55 kDa band is the 53 kDa glycoprotein.

In their 1980 publication [10], Michalak pointed out that the Weber and Osborn SDS-PAGE analysis revealed that the ratio of Coomassie blue-staining intensity of the protein bands at 44 kDa (calsequestrin) and 55 kDa (HACBP) was about 2/1. However, with the Laemmli SDS-PAGE analysis, calsequestrin moved with lower mobility, corresponding to 63 kDa, and importantly, the relative staining density between the calsequestrin band and the 55 kDa band (HACBP) changed to about 5/1. This was important because it confirmed that the HACBP was reasonably abundant (20% of calsequestrin) and should be easily detectable. This result also speaks to the gross impurity of the 55 kDa band, since this Coomassie blue-staining band of 55 kDa identified by MacLennan decreased by more than 50% in intensity when analyzed by the Laemmli SDS-PAGE system. Since this band was purified by chromatography using the detection of proteins of 55 kDa by SDS-PAGE according to Weber and Osborn, the purified protein referred to as the HACBP reported in 1974 by MacLennan’s laboratory must contain multiple proteins.

## 3. Analysis of Rabbit Skeletal Muscle SR by SDS-PAGE

As discussed, one of the significant difficulties in addressing the issue of the identity of the 55 kDa Comassie blue-staining band reported by MacLennan’s laboratory and named the HACBP is that this band comprised several proteins. This was first discovered by Michalak, who noted that the multiple 55 kDa proteins were resolved by ion exchange chromatography of the SR preparation and only detectable by the Laemmli SDS-PAGE procedure [10]. Using the Weber and Ornstein method, they observed a single band at 55 kDa using one-dimensional SDS-PAGE. However, when they employed a two-dimensional separation using Weber and Ornstein SDS-PAGE in one dimension and Laemmli SDS-PAGE in the second dimension, the 55 kDa protein was further resolved into multiple 55 kDA bands. Since the 1974 publication by the MacLennan laboratory used the Weber and Osborn method to identify the 55 kDa band during purification, it is unclear which and how many of the 55 kDa proteins were purified and identified as the HACBP. The fact that this band was, according to Campbell (Figure 2), composed of only the 53 kDa glycoprotein and, according to Michalak [10], composed of the 53 kDa glycoprotein and the HACBP suggests that, at best, the 55 kDa band was composed of multiple distinct proteins or more likely misidentified as the 53 kDa glycoprotein. This indicates that the HACBP identified in 1974 by MacLennan does not exist as a particular protein species.

In 1990, MacLennan’s group used the higher resolution Laemmli SDS-PAGE procedure [11] to reanalyze the distribution of proteins in the skeletal muscle SR (Figure 3) [27]. To aid in the localization of the proteins of the SR, their highly purified SR preparation was subfractionated into longitudinal SR vesicles (LSR) and TC and then analyzed by SDS-PAGE. The major proteins detected within the 66–45,000 dalton region of the LSR were calsequestrin (CaS) as a minor band and the 53 kDa glycoprotein (GP-53) as a major band. They resolved CaS as the prominent band and GP-53 as a minor band of the TC. Conspicuously absent was any mention of the 55 kDa HACBP in these SR fractions. The inability to detect the HACBP was unexpected as Michalak showed the TC to be the site of localization for the HACBP [10]. Furthermore, as discussed, the ratio of the staining intensity of the ATPase to the HACBP was reported to be about 9/1 [10], suggesting that, considering the strong signal by the ATPase band, the HACBP should be easily detected. However, papers that provide an SDS-PAGE analysis of the proteins of the SR, published after the 1974 publication from the MacLennan laboratory [5], have, except for the 1980 Michalak publications [10,28], failed to identify (or discuss) the presence of the 55 kDa HACBP in their analysis [12,27,29,30,31]. In a pivotal review of the Ca^2+^-binding proteins of the SR, Maclennan described in detail the mechanism of Ca^2+^ cycling by the SR; however, the HACBP was not mentioned, nor was the HACBP illustrated in their SR model [32].

Rossi [33] reviewed the work of multiple laboratories that have catalogued and studied the proteins of the SR. They reported a detailed characterization of these proteins. HACBP was not detected in this analysis as an SR protein. Perhaps the most comprehensive analysis of skeletal muscle SR proteins by SDS-PAGE was published by Treves [34]. Their publication, entitled “Minor sarcoplasmic reticulum membrane components that modulate excitation-contraction coupling in striated muscles,” presented a Coomassie blue-stained gradient (5–15%) SDS-PAGE gel analysis of the protein components present in the LSR, in TC, and in the junctional face membrane (JFM) fractions obtained from rabbit skeletal SR (Figure 4).

They reported that the major component of the LSR, constituting approximately 80% of the total proteins present, was the 110 kDa Ca^2+^-ATPase (SERCA). They noted a 22 kDa protein band present in the longitudinal SR fraction that was identified as phospholamban, as well as two glycoproteins of 160 kDa (sarcalumenin) and 53 kDa (53 kDa glycoprotein) that were minor protein constituents of the LSR membrane fraction. The components of the JFM included the abundant components, namely the ryanodine receptor (RyR), calsequestrin (CsQ), histidine-rich Ca^2+^-binding protein, and the triadin(s) (TRISK). Five minor membrane protein components were identified: mitsugumin-29, a 29 kDa membrane protein; junctophilin-1 (72 kDa); Mitsugumin-33 or TRIC-A (trimeric intracellular cation-selective channel), also known as SRP-27 (27 kDa); JP-45 (45 kDa); and junctate (33 kDa). Importantly, their analysis did not detect or mention the HACBP or CRT. This analysis should be viewed from the perspective that according to Michalak, the HACBP comprises about 10% of the Coomassie blue-staining intensity observed for SERCA in rabbit skeletal muscle SR preparations [10].

Raeymaekers used immunodetection of CRT by SDS-PAGE and Western blotting to analyze several SR and ER preparations for CRT. While detecting robust immunoreactivity of CRT in the ER fractions of various tissues, they failed to detect CRT in the SR of skeletal muscle [36]. The CRT immunoreactivity detected in the ER was of molecular weight 63 kDa (Figure 5).

## 4. Analysis of Rabbit Skeletal Muscle SR by Affinity Techniques

Stains-All has been used to detect Ca^2+^-binding proteins in general and specifically to identify and characterize the Ca^2+^-binding proteins of skeletal muscle SR [37,38,39]. This dye, which stains Ca^2+^-binding proteins deep blue to violet, has been shown to bind to Ca^2+^-binding sites of Ca^2+^-binding proteins and to be a sensitive method to detect Ca^2+^-binding proteins. Nomura [39] compared the sensitivity of Stains-All with Coomassie blue and found that Stains-All and Coomassie blue stained CRT with equal intensity, thus illustrating the high sensitivity of Stains-All. As discussed, in 1983, Campbell’s group analyzed the protein composition of skeletal muscle fractions obtained during the purification of the SR. They used the same methodology reported earlier by MacLennan’s group to isolate the skeletal muscle SR [38]. Within the 66–45 kDa region, they identified the presence of calsequestrin and GP-53 by staining SDS-PAGE gels with Coomassie blue or Stains-All (Figure 6). The presence of the HACBP was not detected in the crude skeletal muscle fractions or the purified SR. Interestingly, earlier papers by Campbell and MacLennan in 1981 and 1982 that provided Laemmli SDS-PAGE analysis of skeletal muscle SR proteins also failed to either detect or discuss the 55 kDa HACBP in their SR preparation but cleanly resolved the GP-53 from calrequestrin [30,31].

Damiani and Margreth also extensively analyzed the proteins of skeletal muscle SR. They prepared SR membrane subfractions, including highly purified longitudinal tubules, junctional transverse tubules, or junctional free membrane [40]. Using the Stains-All procedure, they identified calsequestrin within the 66–45 kDa region but were also unable to detect the HACBP (Figure 7).

The ^45^Ca^2+^ overlay procedure is a sensitive method for detecting CRT and other Ca^2+^-binding proteins [14]. Macer and Koch [14] directly compared the Ca^2+^-binding proteins of the ER and SR by ^45^Ca^2+^ autoradiography. They observed that CRT (labelled Car) is detected in the ER but not in the SR (Figure 8). They showed that the main Ca^2+^-binding protein of the SR is calsequestrin (labelled Cas).

In addition to Stains-All, Damiani and Margreth also utilized SDS-PAGE with the ^45^Ca^2+^ overlay technique to detect the major Ca^2+^-binding proteins of skeletal SR fractions (Figure 9). The SR membranes were fractionated into TC (Figure 9, lanes 1 and 3) and LSR (Figure 9, lanes 2 and 4). Within the 66–45- kDa region, they observed a strong signal for calsequestrin. The HACBP or CRT was neither detected nor mentioned in their publication.

Similarly, Picello reported that the ^65^Zn^2+^ overlay technique detected several ^65^Zn^2+^-binding SR proteins, including the histidine-rich Ca^2+^-binding protein (HRC), Ryanodine receptor (RyR) protein monomer, and Ca^2+^-ATPase protein [41]. CRT, a well-known ^65^Zn^2+^-binding protein [19,42], was not detected in the TC or JFM fractions.

Hofman [43] used Stains-All and ^45^Ca^2+^ overlay to detect the major Ca^2+^-binding proteins of skeletal muscle SR (Figure 10). They observed that the major band detected by Stains-All was calsequestrin (Figure 10, lane 1), and minor staining bands were shown to be sarcalumenin. The ^45^Ca^2+^ overlay analysis identified calsequestrin as the major band (Figure 10, lane 3). The HACBP or CRT was not detected or discussed in their publication.

In 1989, MacLennan’s group reported a ^45^Ca^2+^ overlay analysis of the SR [29]. They used highly purified preparations of longitudinal SR (Figure 11, lane 2) and terminal cisternae (Figure 11, lane 3) and reported that the major Ca^2+^-binding protein of skeletal muscle SR, detected by ^45^Ca^2+^ overlay analysis, was calsequestrin (Figure 11-CaS), the 160 kDa glycoprotein (GP-160), and the 170- and 200 kDa Ca^2+^ binding proteins. Calsequestrin, shown to be present in the terminal cisternae, displayed a robust signal. Still, any other bands below calsequestrin that would correspond to the HACBP or CRT were conspicuously absent. They also analyzed all SR proteins using Coomassie blue staining of the purified SR fraction. They reported the positive staining of the 53 kDa glycoprotein, calsequestrin, the Ca^2+^-ATPase (ATPase), the 160 kDa glycoprotein (GP-160), and proteins with apparent molecular masses of 92, 100, and 170 kDa. A protein band corresponding to the 55 kDa HACBP was not detected. Surprisingly, their publication failed to describe or discuss the HACBP or CRT.

Therefore, the compilation of the publications suggests that after MacLennan reported the HACBP in 1973/1974 [4,5], evidence of the presence of HACBP in skeletal muscle SR was only reported for publications from the Michalak laboratory in 1980. In subsequent publications from the MacLennan laboratory in 1981, 1983, 1983, 1989, and 1990 [12,27,29,30,31], the HACBP was not mentioned nor mentioned in their review of the Ca^2+^-binding proteins of the SR [32]. Interestingly, the disappearance of the HACBP seems to have occurred at the same time as the appearance of the 53 kDa glycoprotein on Coomassie blue-stained SDS-PAGE [31].

## 5. Identity of the 55 kDa Protein of the Rabbit Muscle SR

Rigorous analysis of the SR proteins by laboratories specializing in the characterization of SR proteins has failed to demonstrate the presence of the HACBP or CRT in the rabbit muscle SR preparations. We, therefore, conclude that the HACBP does not exist in the skeletal muscle SR as a protein species. We, therefore, propose that the burden of proof has been met that the CRT was not identified initially as the HACBP of rabbit skeletal muscle SR. Thus, the correct historical record is that CRT was initially discovered as a liver Ca^2+^-binding protein of the endoplasmic reticulin [17,18], extensively characterized [14], and named calregulin. Its limited amino acid sequence was published [16] and later completed by Koch [13] and confirmed by Michalak [15].

It is interesting to speculate as to the identity of the illusive 55 kDa SR protein(s) initially reported by MacLennan in 1974. As discussed, in 1980, Michalak showed that the 55 kDa protein band, resolved by SDS-PAGE and referred to as the HACBP, contained multiple 55 kDa proteins [10]. However, in 1989, Michalak claimed to sequence the HACBP [15]. This was accomplished using a freshly prepared polyclonal goat antibody purified against CRT isolated from rabbit skeletal muscle homogenates by ammonium sulfate precipitation and ion exchange chromatography. The original HACBP antibody was not used. The Michalak antibody was used to screen λgt11 cDNA expression libraries constructed from poly(A)+ RNA of rabbit slow-twitch (soleus) and fast-twitch (psoas) muscle [15]. As pointed out by Opas [44,45], CRT is mainly present in the fibroblasts and endothelial cells that are present in this muscle preparation. Therefore, since the antibody used to screen the library was not affinity purified against the 55 kDa SR proteins (HACBP), it cannot be stated that Michalak cloned the HACBP, merely that his laboratory produced an antibody against CRT purified from skeletal muscle tissue homogenates and used this antibody to clone and sequence skeletal muscle CRT. This sequence was in excellent agreement with the original reported sequence for CRT [13].

Michalak used the same polyclonal antibody that was used to screen the λgt11 cDNA expression libraries for CRT, and this time, he reported the sequencing of another 55 kDa protein in skeletal muscle SR. Based on sequence analysis, he called the 55 kDa protein the multifunctional thyroid hormone-binding protein of skeletal muscle [46]. Subsequently, this protein was identified as protein disulfide isomerase (PDI), an acidic Ca^2+^-binding protein of the ER but not SR [47,48]. The fact that Michalak isolated and sequenced this protein, which is now known to be a KDEL-containing ER protein, lends credence to the possibility that the affinity purification of the antibody against a partially purified skeletal muscle homogenate that would contain SR from the skeletal muscle and ER from the endothelial cells and fibroblasts resulted in an ER-reactive antibody. Importantly, when Michalak tested this antibody against authentic bovine liver calregulin (CRT) obtained from our laboratory, it failed to react against our protein [49].

Secondly, calsequestrin is present in muscle as two isoforms, and the calsequestrin-2 isoform has a molecular weight of 55 kDa and is present in slow-twitch skeletal muscle fibers. It is possible that the HACBP band contained calsequestrin-2 or a proteolytic fragment of calsequestrin. It is unclear why the 55 kDa protein disappeared from the analysis of the SR proteins after 1980. Michalak reported that the binding of ^125^I-concanavalin A by a protein of about 53,000 daltons by Laemmli SDS-PAGE and about 55,000 daltons by Weber and Osborn SDS-PAGE. The original SR analysis that identified the HACBP, did not resolve the 53 kDa glycoprotein. Therefore, this protein was likely the major protein present in the original 1974 HACBP band. Since multiple laboratories have failed to detect the presence of CRT in SR preparations, the 55 kDa HACBP cannot be CRT. Since the HACBP was likely a hodgepodge of several proteins, speculation about the identities of the HACBP identified in 1974 is moot. We propose that considering, as discussed, that expert analysis by laboratories who actively study the SR has failed to identify HACBP/CRT in the SR, the discovery paper for CRT should be the publication that first identified and purified CRT from bovine liver [17] and identified CRT as a Ca^2+^-binding, ER protein [18]. The discovery of CRT is discussed in Section 5.

Finally, it is interesting to note that the early skeletal muscle preparations did not utilize proteolytic inhibitors during the isolation of SR fractions. We were only able to find mention of the use of the inhibitor PMSF (a serine protease inhibitor) in the 1983 Campbell analysis of SR proteins [12], in which the HACBP was not detected.

As shown by Michalak, proteolysis of the SR preparation results in the enrichment of the 55 kDa protein [10] (Figure 12A). This experiment was reproduced by Zorzato, who also showed that incubation with the SR preparation with trypsin resulted in the appearance of a 55 kDa Coomassie blue-staining band in SDS-PAGE (Figure 12B) [50]. Levitsky et al. [51] also showed that tryptic digestion of the SR Ca^2+^-ATPase generates a Coomassie blue-staining band in SDS-PAGE of 55 kDa. The enrichment of the 55 kDa Coomassie blue-staining bands after proteolytic digestion of the SR preparation in vitro observed by several laboratories is consistent with the possibility that the 55 kDa band is artifactually generated by the proteolysis of other protein(s) that occurs during the preparation of the SR fractions.

## 6. From Calregulin to Calreticulin—The Discovery and History of Calreticulin

Sidney Ringer initially reported that Ca^2+^ played an important role as a signaling molecule. He demonstrated the importance of this ion in muscle contraction [52]. Subsequently, Heilbrunn reported that the addition of Ca^2+^ caused the immediate and pronounced shortening of frog muscle [53]. Ebashi later reported that the receptor that mediated Ca^2+^-dependent regulation of muscle contraction was a protein that bound Ca^2+^, which he named troponin. Therefore, troponin was the first intracellular Ca^2+^-binding protein identified [54,55]. It was subsequently demonstrated that a subunit of troponin, namely troponin-C, was a Ca^2+^-binding protein and that the binding of Ca^2+^ to troponin-C induced a conformational change in troponin that resulted in the activation of actomyosin ATPase [56]. In 1970, Dr. Rasmussen proposed that Ca^2+^ acted as a second messenger similar to that of c-AMP [57]. At this time c-AMP was known to bind to a specific binding pocket in the c-AMP-dependent protein kinase. This presented the possibility that proteins capable of interacting with Ca^2+^ and modulating the regulatory role of Ca^2+^ might exist. The search was on for proteins that might act as a receptor for this second messenger, i.e., Ca^2+^-binding proteins. In 1973, a Ca^2+^-binding protein was purified, and this protein was subsequently called calmodulin [58]. At that time, calmodulin was thought to only regulate cyclic nucleotide phosphodiesterase and thereby provide a link between the Ca^2+^ and cAMP second messenger systems [59]. However, it was soon demonstrated that calmodulin was highly conserved and present in a wide range of organisms, which led to the conclusion that calmodulin was ubiquitously distributed and fundamentally important to the second messenger system [60,61]. Calmodulin was subsequently shown to regulate a large number of proteins and processes [62].

We investigated the possibility that cells might utilize other Ca^2+^-binding proteins, in addition to calmodulin, to modulate the Ca^2+^ second messenger system. We prepared the 100,000× *g* supernatants of a variety of bovine tissues, chromatographed these extracts on ion-exchange columns, and assayed the resultant fractions for Ca^2+^-binding activity using the Chelex competitive ^45^Ca^2+^-binding assay [63]. We observed that when liver homogenates were analyzed by our procedure, the major peak of Ca^2+^-binding activity was distinct from calmodulin. The Ca^2+^-binding protein that was responsible for the Ca^2+^-binding activity peak was purified, and a 63 kDa protein characterized. The amino acid composition suggested that the protein was negatively charged, and analysis of the Ca^2+^-binding activity revealed a single high-affinity Ca^2+^ bound with half-maximal binding at 0.1 μM in the presence of 3 mM Mg^2+^ and 50 mM KCl [17]. We then extensively characterized the physical properties of this protein, called calregulin (CAB-63), and demonstrated its colocalization with the ER of fibroblasts [18]. Subsequently, we demonstrated using our rabbit anti-calregulin antibody that calregulin is present in a wide variety of bovine tissues [20]. We also reported that calregulin contains distinct and specific ligand-binding sites for Ca^2+^ and Zn^2+^. Our detailed analysis revealed that in the presence of 3.0 mM MgCl_2_ and 150 mM KCl, calregulin had a single binding site for Ca^2+^ with an apparent dissociation constant of 0.05 μM and 14 binding sites for Zn^2+^ with an apparent Kd (Zn^2+^) of 310 μM. Ca^2+^ binding to calregulin resulted in a 5% increase in intrinsic fluorescence intensity and a 2–3-nm blue shift in emission maximum. In contrast, Zn^2+^ binding to calregulin causes an increase of about 250% in intrinsic fluorescence intensity and a red shift in the emission maximum of about 11 nm. We concluded that Ca^2+^ binding resulted in the movement of tryptophan away from the solvent, and Zn^2+^ binding caused a movement of tryptophan into the solvent and the exposure of a hydrophobic domain [19]. Our studies of the purification and characterization of calreticulin were summarized in our 1987 methodology review [21].

We also performed a detailed study of the structure and evolution of calregulin, using calregulin from bovine, chicken, and rabbit livers. This 1987 publication was the first report of the partial amino acid sequence of CRT [16]. We also reported that in the presence of 3.0 mM MgCl2 and 150 mM KCl, all the calregulins bound 1.0 mol of Ca^2+^/mol of protein, with apparent Kd values of 0.05 μM (bovine) and 0.03 μM (chicken and rabbit).

## 7. Conclusions

After an extensive analysis of the scientific literature, we conclude that the 1974 publication by the MacLennan laboratory that identified the 55 kDa Coomassie blue-staining band in rabbit skeletal muscle SR preparations as the HACBP cannot be reproduced (Figure 1). We make this statement based on the evidence presented in this review. Perhaps the most convincing data were presented by Campbell in 1983, who used the identical SR preparation to the 1974 publication and showed that the 55 kDa Coomassie blue-staining band, referred to as the HACBP by MacLennan, resolved into a single protein by 2D SDS-PAGE, which they identified as the 53 kDa glycoprotein (Figure 2). Interestingly, the detection of the HACBP or discussion of the HACBP was abandoned by the MacLennan laboratory in subsequent publications involving SDS-PAGE analysis of the proteins of skeletal muscle SR in 1981 [31], 1982 [30], 1983 [12], 1989 [29], and 1990 [27]. The HACBP was also not mentioned in an extensive review of the SR proteins published by the MacLennan laboratory in 2002 [32]. The last publication from the MacLennan laboratory that described the HACBP was the publication in 1980 by Michalak [10]. This publication clearly demonstrated that the HACBP was a hodgepodge of several 55 kDa proteins. Of significance, the 55 kDa protein band, claimed to be the HACBP, was represented as being in reasonable abundance; i.e., according to Michalak, the relative staining density between the band containing calsequestrin and the band containing the HACBP was about 5/1. This suggests that the detection of this protein by SDS-PAGE should be relatively easy. Furthermore, the demonstration by Michalak and other laboratories (Figure 12) that treatment of SR preparations with proteases increases the intensity of the 55 kDa protein Coomassie blue-staining band lends credence to the possibility that the HACBP observed by Michalak in 1980 contained a proteolytic artefact in addition to the 53 kDa glycoprotein.

Other laboratories, experts in the analysis of skeletal muscle SR proteins, have also failed to identify the HACBP in SR preparations. Collectively, the data from these laboratories and the post-1980 publications from the MacLennan laboratory establish that the HACBP fails the test of reproducibility. Since the experts in the SR field have failed to identify HACBP in skeletal muscle SR, the HACBP and CRT cannot be identical proteins. Therefore, the claim that the CRT was originally identified in skeletal muscle SR as the HACBP is a non sequitur. Since the antibody used for the sequencing of the HACBP was generated against a protein purified from skeletal muscle homogenates, and not skeletal muscle SR, there is no link between the HACBP and the sequencing of CRT. It is therefore more correct to state that the antibody was affinity-purified against CRT and then used to sequence this protein. This antibody was also used to clone PDI, an established ER protein, which suggests that the skeletal muscle preparation used to affinity-purify the CRT antibody contained ER from other cell types present in the muscle tissue.

Therefore, CRT was actually discovered as a novel hepatic Ca^2+^-binding protein that was purified in 1984 [17], extensively characterized in 1985 [18], and partially sequenced in 1987 [16]. This protein was called calregulin.

## Figures and Tables

**Figure 1 biomolecules-14-00866-f001:**
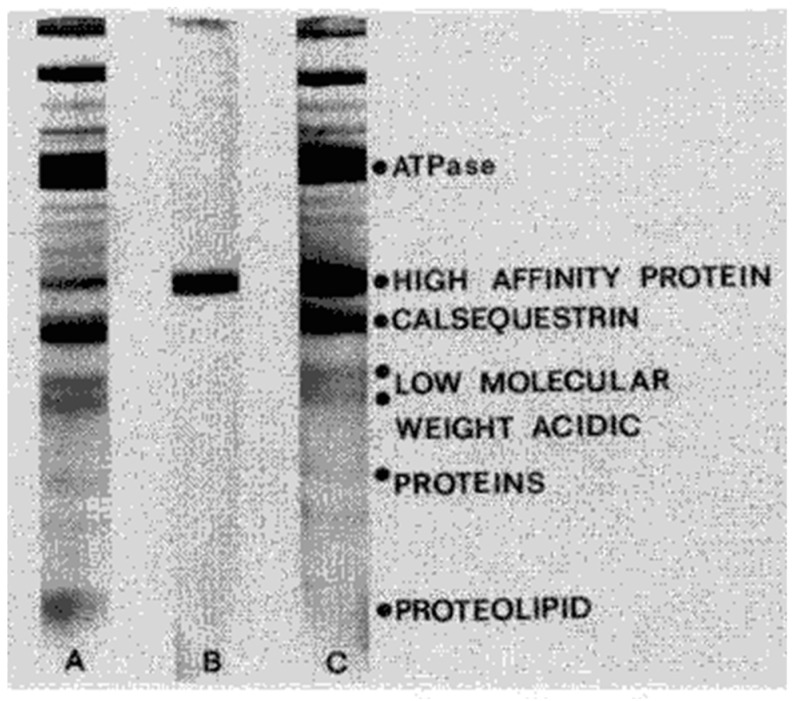
Resolution of SR proteins by SDS-PAGE. SDS-PAGE analysis of (**A**) purified rabbit skeletal muscle SR; (**B**) purified HACBP; (**C**) a mixture of (**A**,**B**). MacLennan’s laboratory initially identified the HACBP and reported that the HACBP was an important SR Ca^2+^-binding protein. This figure, reproduced from the original report, shows the presence of the HACBP in skeletal muscle SR resolved according to the SDS-PAGE procedure of Weber and Osborn [6]. This research was originally published in the [5]. Reproduced with permission from [5]; published by Elsevier, 1974. © 1974 American Society for Biochemistry and Molecular Biology (ASBMB).

**Figure 2 biomolecules-14-00866-f002:**
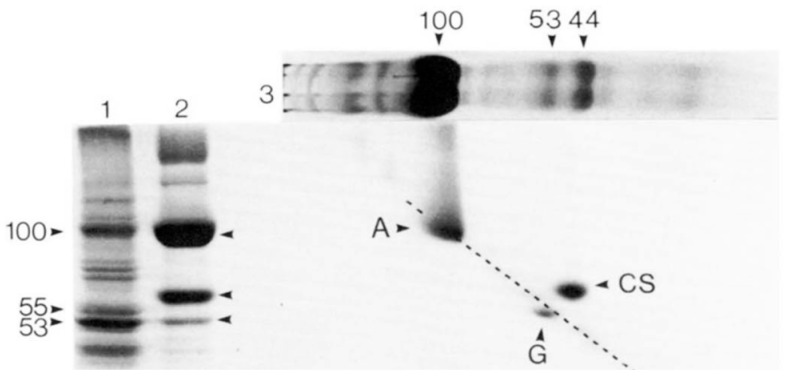
Two-dimensional gel electrophoresis of skeletal and cardiac muscle SR proteins. The SR from rabbit skeletal muscle was subjected to two-dimensional gel electrophoresis and stained with Coomassie blue. Skeletal muscle calsequestrin (CS) fell above the diagonal, since its mobility differed in the two gel systems. By contrast, the 53 kDa glycoprotein (G) fell slightly below the diagonal. Reference gels show cardiac SR (Lane 1) and skeletal SR (Lane 2) analyzed according to Laemmli SDS-PAGE. The skeletal SR was analyzed by Weber and Osborn SDS-PAGE in the first dimension and Laemlli in the second dimension (Lane 3). This figure highlights that the HACBP initially detected by the MacLennan laboratory (Figure 1, lane A) is not present in SR preparations subsequently reported by this laboratory (Figure 2, lanes 2,3) but that analysis of the HACBP band by two-dimensional PAGE identifies this band as the 53 kDa glycoprotein (G). A, Ca^2+^-ATPase. Reproduced with permission from Ref. [12]; published by Elsevier, 1974. © 1983 American Society for Biochemistry and Molecular Biology (ASBMB).

**Figure 3 biomolecules-14-00866-f003:**
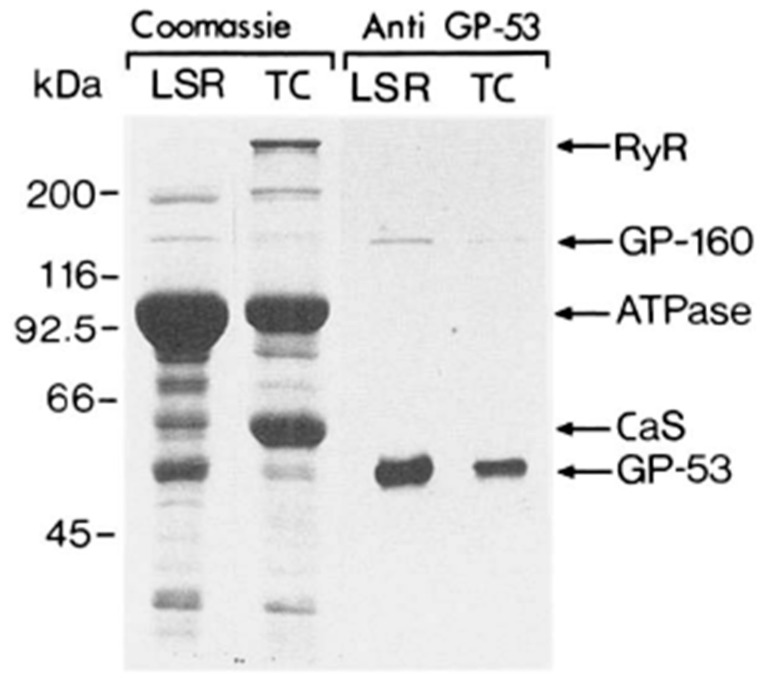
Analysis of proteins in isolated SR subfractions. The longitudinal SR (LSR) and the terminal cisternae (lanes TC) were analyzed. (**Left**), Coomassie-stained gel; (**Right**), staining with anti-53 kDa glycoprotein monoclonal antibody. Note the 53 kDa glycoprotein, i.e., the major band that migrates to a position below calsequestrin. In contrast to their 1974 study (Figure 1), this analysis of the SR fractions reported in 1990 showed that the HACBP was not detected in the SR fractions. This research was originally published in the [27]. Reproduced with permission from [27]; published by Elsevier, 1990. © 1990 American Society for Biochemistry and Molecular Biology (ASBMB).

**Figure 4 biomolecules-14-00866-f004:**
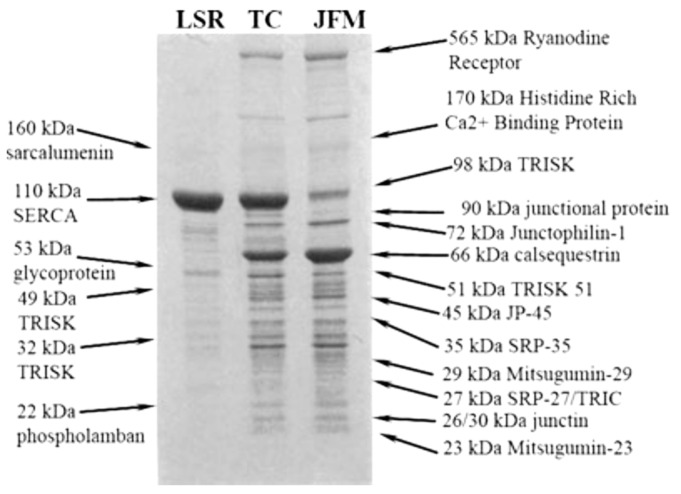
Analysis of skeletal muscle SR proteins by Coomassie blue staining. This figure presents an exhaustive and detailed analysis of major and minor SR proteins detected by Coomassie blue staining of a 5–15% gradient SDS-PAGE gel [35]. This analysis did not detect CRT or the HACBP, nor was it described as an SR protein by these authors. A Coomassie blue-staining band is not observed between calsequestrin (66 kDa) and the 53 kDa glycoprotein. LSR, longitudinal SR; TC, terminal cisternae; JFM, junctional-face membrane. This research was initially published by [34]. Reproduced with permission from [35]; published by Elsevier, 1986. © 1990 American Society for Biochemistry and Molecular Biology (ASBMB).

**Figure 5 biomolecules-14-00866-f005:**
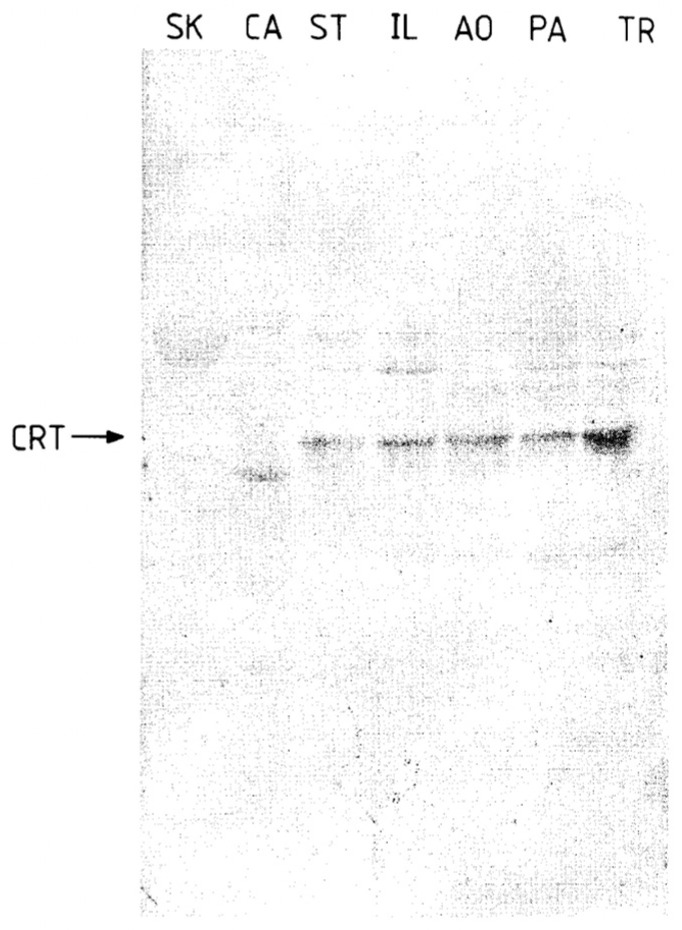
Immunological identification of CRT in SR and ER fractions of smooth muscle. SR and ER membrane proteins were separated by SDS-PAGE on 7.5% polyacrylamide gels, transferred to Immobilon-P filters, and incubated with anti-CRT antiserum at 1:200 dilution. A total of 40 μg of protein was loaded on each lane: pig skeletal muscle SR (lane SK), cardiac muscle SR (lane CA), smooth muscle ER from stomach (ST), ileum (IL), pulmonary artery (PA), aorta (AO), and trachea (TR). The arrow indicates CRT, Mr, 63 kD. Reproduced from [36]. Reproduced with permission from [36]; published by Elsevier, 1993. © 1993 Elsevier Ltd.

**Figure 6 biomolecules-14-00866-f006:**
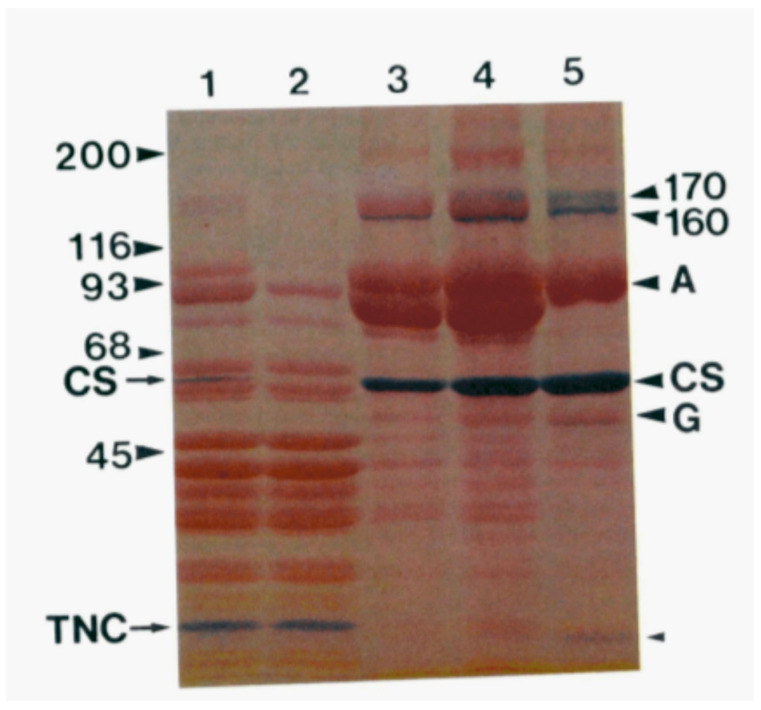
Analysis of SR proteins by Stains-All. Coomassie blue and Stains-All staining of skeletal muscle proteins. Fractions in the purification of SR vesicles from rabbit skeletal muscle were analyzed by SDS-PAGE and stained with Stains-All. Lane 1, supernatant from rabbit skeletal muscle homogenate following centrifugation at 10,000× *g* for 20 min; lane 2, supernatant from rabbit skeletal muscle homogenate following centrifugation at 50,000× *g* for 1 h; lane 3, pellet obtained from 50,000× *g* centrifugation; lane 4, supernatant following 7000× *g* centrifugation; lane 5, KC1-washed SR vesicles. The Stains-All-stained gel contained 200 μg of protein in each lane. A, ATPase (105,000 Da); CS, calsequestrin (63,000 Da); G, 53 kDa glycoprotein; 160, 160,000-Da glycoprotein; 170, 170,000-Da protein; TNC, troponin C. This figure shows that although calsequestrin is easily detected in SR by Stains-All staining of SDS-PAGE gels, CRT is not detected by this stain in the purified SR (lane 5) or in any of the other fractions isolated from skeletal muscle. This research was originally published in the [38]. Reproduced with permission from [38]; published by Elsevier, 1983. © 1983 American Society for Biochemistry and Molecular Biology (ASBMB).

**Figure 7 biomolecules-14-00866-f007:**
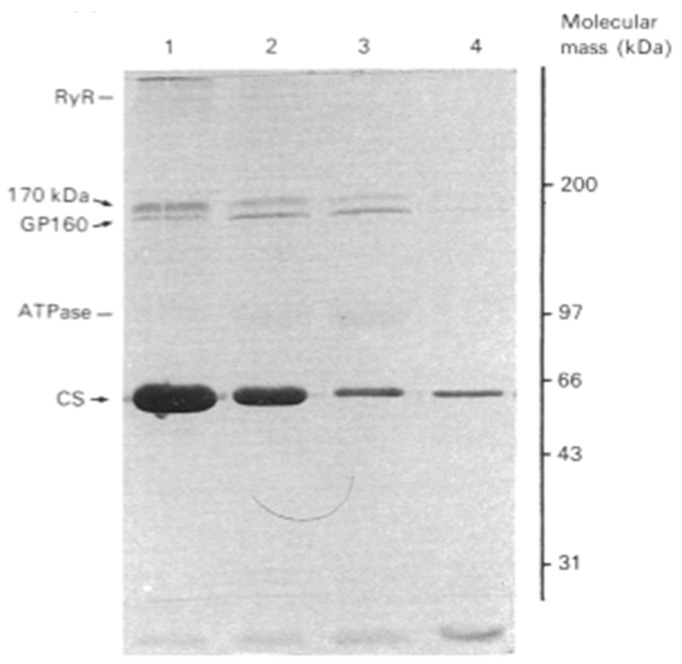
Analysis of SR fractions by Stains-All. About 50 μg of protein was loaded per lane. Protein bands indicated by arrows were stained blue. SR subfractions are numbered from the bottom to top of the gradient. Lanes: 1, junctional TC; 2, intermediate fraction; 3, fraction enriched in LSR; 4, light fraction. Abbreviation: GP160, 160 kDa Ca^2+^-binding glycoprotein (sarcalumenin). This figure provides a second independent analysis of the proteins of SR identified by Stains-All. Consistent with the data presented in Figure 6, calsequestrin but not CRT is identified in skeletal muscle SR fractions. This figure was from research originally published by [40]. Reproduced with permission from [40]; published by Biochemical Society (Great Britain), 1906. © Portland Press, Ltd.

**Figure 8 biomolecules-14-00866-f008:**
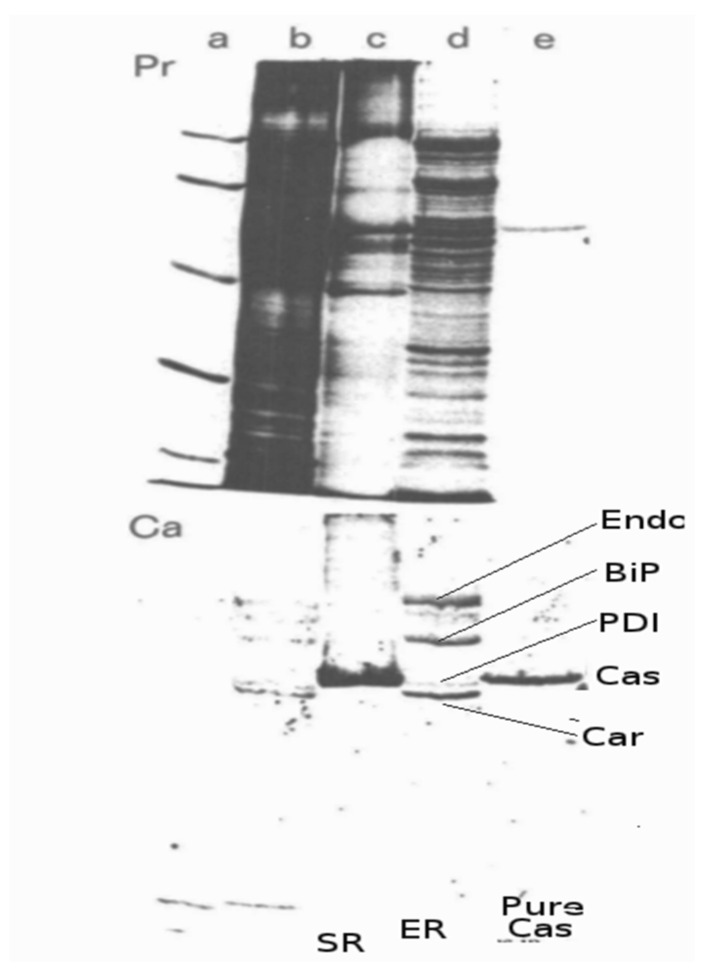
Comparison of ER and SR proteins detected by ^45^Ca^2+^ overlay. Samples were prepared and analyzed as described by Macer and Koch, 1988 [14]. Each panel shows the protein (Pr) and ^45^Ca^2+^ autoradiograph (Ca) from the same sample. Lanes (**a**) protein standards (from top): 95 K, 67 K, 45 K, 30 K, 17 K; (**b**) whole cell lysate from MOPC-315; (**c**) SR; (**d**) ER reticuloplasm from MOPC-315 cells; (**e**) purified calsequestrin (CaS). Endo, endoplasmin; BiP, immunoglobulin heavy chain-binding protein; PDI, protein disulfide isomerase; Car, calreticulin. This research was originally published as [14].

**Figure 9 biomolecules-14-00866-f009:**
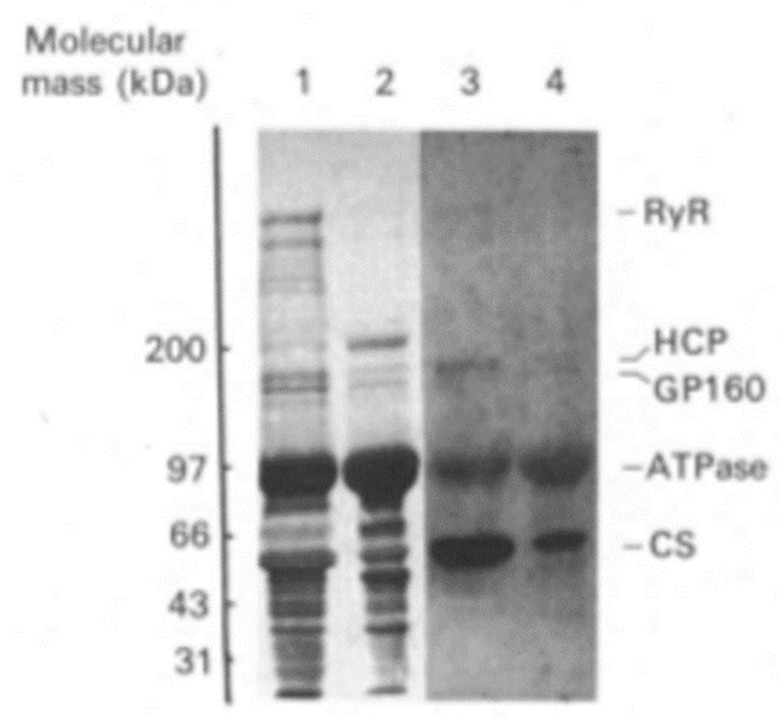
Identification of SR proteins by ^45^Ca^2+^ autoradiography. Proteins were resolved by 5–10% gradient PAGE and transferred to nitrocellulose. Blots were stained with Ponceau Red (lanes 1 and 2) and then incubated with ^45^Ca^2+^. ^45^Ca^2+^-labeled proteins (lanes 3 and 4) were detected by autoradiography after a 7-day exposure. This figure shows that calsequestrin but not CRT is detected in SR fractions by ^45^Ca^2+^ autoradiography. This research was originally published as [40]. Reproduced with permission from [40]; published by Biochemical Society (Great Britain), 1906. © Portland Press, Ltd.

**Figure 10 biomolecules-14-00866-f010:**
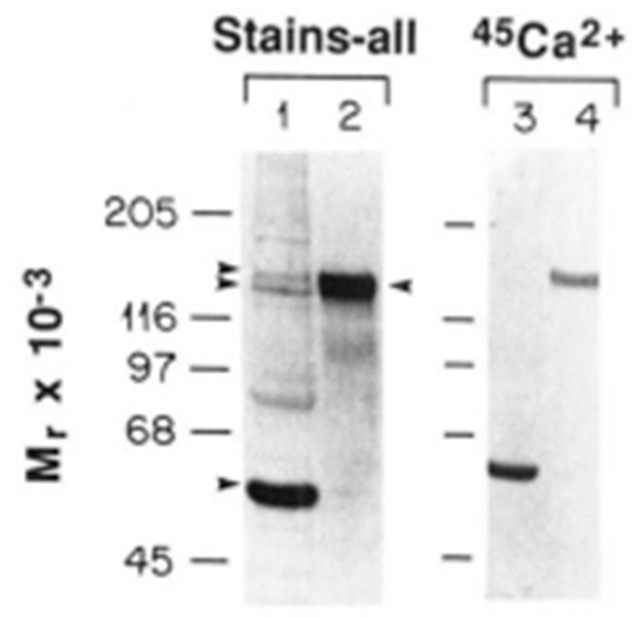
Detection of SR proteins by Stains-All and ^45^Ca^2+^ autoradiography. Solubilized rabbit muscle membranes (130 μg) (lanes 1 and 3) and the purified 165 kDa protein (5 μg) (lanes 2 and 4) were analyzed by SDS-PAGE and either stained with Stains-All or transferred to nitrocellulose and subjected to analysis by ^45^Ca^2+^ autoradiography. The prominent Stains-All binding protein and ^45^Ca^2+^-binding protein at 63 kDa co-migrated with authentic rabbit calsequestrin. This figure shows that in contrast to calsequestrin, CRT is not detected by Stains-All staining or ^45^Ca^2+^ autoradiography of skeletal muscle fractions. This research was originally published in the [43]. Reproduced with permission from [43]; published by Elsevier, 1989. © 1989 American Society for Biochemistry and Molecular Biology (ASBMB).

**Figure 11 biomolecules-14-00866-f011:**
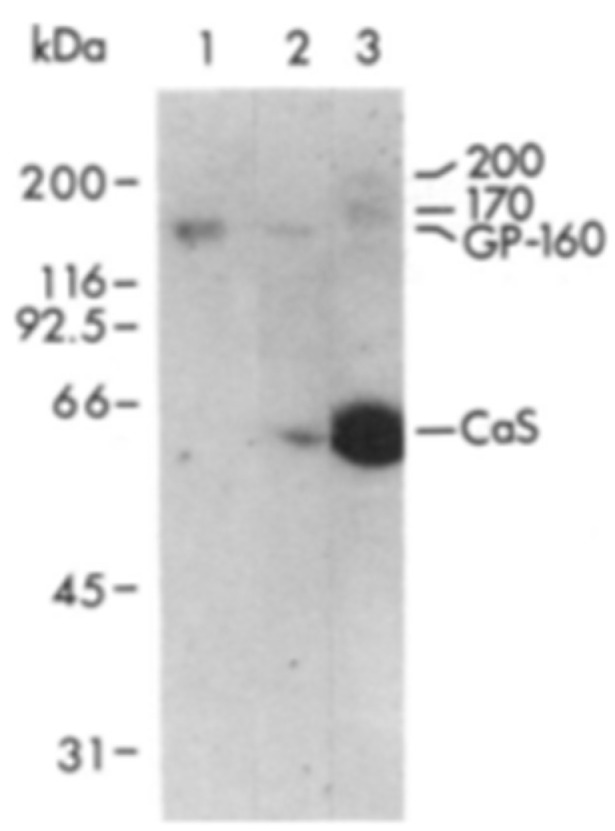
Detection of SR proteins by ^45^Ca^2+^ autoradiography. Identification of ^45^Ca^2+^-binding proteins from SR. Purified 160 kDa glycoprotein (lane 1), longitudinal SR (lane 2), and terminal cisternae (lane 3) were subjected to SDS-PAGE and transferred to a nitrocellulose membrane and incubated with ^45^Ca^2+^. The panel shows an autoradiograph of the membrane. Molecular masses of standard proteins are given on the left. Positions of calsequestrin (CaS), the 160 kDa glycoprotein (GP-160), and the 170 and 200 kDa Ca^2+^-binding proteins are indicated. This figure, generated by the MacLennan laboratory, shows that calsequestrin but not the HACBP (originally reported by MacLennan as a major Ca^2+^-binding protein of skeletal SR; Figure 1) nor CRT (reported by Michalak’s laboratory to be the HACBP of skeletal muscle) is detected in skeletal muscle SR preparations. This figure was originally published as [29]. Reproduced from [29]; published by NATIONAL ACADEMY OF SCIENCE, 1989. © the Authors.

**Figure 12 biomolecules-14-00866-f012:**
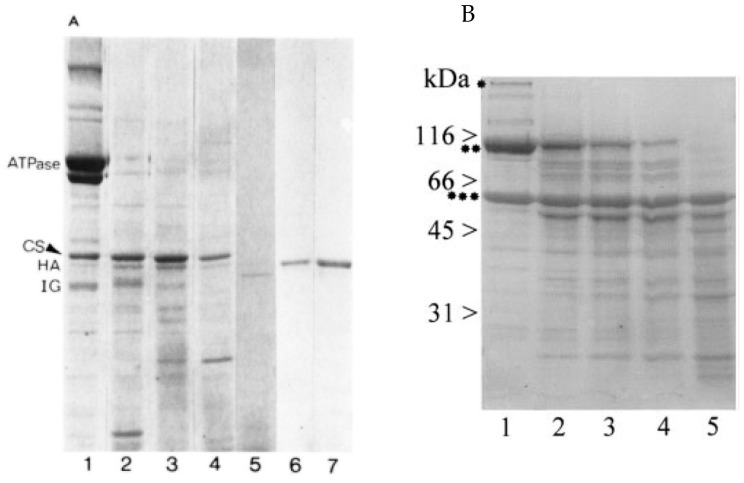
Digestion of SR vesicles with proteolytic enzymes. (**A**) SR vesicles (10 mg/mL) were digested with trypsin, pronase, or papain in the presence of 100 mM KCI, 5 mM CaC12, and 10 mM Tris-HCI, pH 7.5 (9). The protein pattern of the digested samples was analyzed using SDS-gel electrophoresis (7.5% polyacrylamide) according to Laemmli’s method (23). Forty micrograms of protein were applied on gels. Lane 1, intact (original) vesicles; lane 2, vesicles digested with trypsin for 30 min; lane 3,vesicles digested with pronase for 60 min; lane 4, vesicles digested with papain for 30 min; lane 5, vesicles dissolved in 1% Triton X-100 digested with pronase for 30 min; lane 6, HACBP calcium-binding protein isolated from vesicles digested with trypsin for 15 min in the presence of 1 M sucrose; lane 7, HACBP isolated from intact vesicles. CS, calsequestrin; HA, high-affinity calcium-binding protein; IC, intrinsic glycoprotein. Reproduced with permission from [10]; published by Elsevier, 1980. © 1989 American Society for Biochemistry and Molecular Biology (ASBMB). (**B**) SDS/PAGE of rabbit skeletal muscle TC vesicles after mild proteolytic digestion with increasing trypsin/TC ratios. Lanes 1–5 show samples digested with protein/trypsin ratios of 4000:1, 2000:1, 1000:1, 500:1, and 250:1, respectively. Proteins present in lanes 1–5 were stained with Coomassie Brilliant Blue. *, RYR; **, CaATPase; ***, calsequestrin. Reproduced from [50]. This figure shows the generation of the 55 kDa protein by proteolytic digestion of the SR preparation.

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
