# Peer review of "Calreticulin—Enigmatic Discovery"

_biomolecules, 2024, doi:10.3390/biom14070866_

Round 1

Reviewer 1 Report

Comments and Suggestions for Authors

In this manuscript, authors mentioned details of research journey of CRT, with extensive evidence and analysis lead to the conclusion in contract to the previous and proposed new functions of CRT. Here are my comments authors should consider before publishing this.

Major:

1.     This review focuses on many literatures published around 1980, for Ca-binding protein, like calmodulin and the similar lanmodulin have rapidly been researched these days, any recent related research or evidence of CRT?

2.     At beginning, could authors try to add a sketch figure to describe the protein and its potential functions? Especially authors try to propose a new function of CRT.

Minor:

1.     Some typos like line 368, 719. there could be more I did not identify, authors please proof reading.

2.     Figures are all from pervious aged publications, authors could try to make them fit, like Figure 12, the A part could be bigger.

Author Response

Major:

  1.     This review focuses on many literatures published around 1980, for Ca-binding protein, like calmodulin and the similar lanmodulin have rapidly been researched these days, any recent related research or evidence of CRT?

The most recent studies of the skeletal muscle SR (2012 the most recent) have failed to detect or discuss CRT because it is not present in SR, despite claims in 1974 and 1980 to the otherwise. There are no current papers that examine the function of CRT in the skeletal muscle SR.

  1.     At the beginning, could authors try to add a sketch figure to describe the protein and its potential functions? Especially authors try to propose a new function of CRT.

Yes, we have included a summary diagram as a graphical abstract, as you requested. It highlights the new function we have discovered and possible ramifications of that discovery.

Minor:

  1.     Some typos like line 368, 719. there could be more I did not identify, authors please proof reading.--fixed. The spelling of analyzed is correct. We have used Grammerly to proof the manuscript.

  1.     Figures are all from pervious aged publications, authors could try to make them fit, like Figure 12, the A part could be bigger.--this has been fixed. I have repositioned the figures as per your request.

Reviewer 2 Report

Comments and Suggestions for Authors

The manuscript entitled, “Calreticulin, Enigmatic Discovery and Extracellular Function” is a highly unusual article focused primarily on defining who deserves the credit for discovering calreticulin, with a secondary focus on whether or not calreticulin is an ER/SR protein vs. an extracellular protein. I am being asked as a reviewer to assess events that occurred between 40 and 50 years ago. This is not a reasonable task and I am not very comfortable with it. While David MacLennan has died, that is not the case for Marek Michalak or many of the other investigators referenced in this study. I cannot make a recommendation for or against publication, but would like to state the following.

Major Comments:

1.    Given that calreticulin is only marginally expressed in adult muscle, it does seem possible that Dr. Michalak came to the right conclusions for all of the wrong reasons, as it is unlikely that they would have been able to find such a strong calreticulin band in rabbit skeletal muscle.

2.    Calreticulin is an ER protein. I do not find the claims that it is a secreted protein in skeletal muscle only to be credible. Moreover, to the best of my knowledge, the claim that KDEL cannot be read in sarcoplasmic reticulum to be false; KDEL receptors are neither in the ER or the SR. This does not mean that one never finds calreticulin outside of cells. There are many cellular components found in the extracellular milieu.

Author Response

eviewer 2

Major Comments:

The manuscript entitled, “Calreticulin, Enigmatic Discovery and Extracellular Function” is a highly unusual article focused primarily on defining who deserves the credit for discovering calreticulin, with a secondary focus on whether or not calreticulin is an ER/SR protein vs. an extracellular protein. I am being asked as a reviewer to assess events that occurred between 40 and 50 years ago. This is not a reasonable task and I am not very comfortable with it. While David MacLennan has died, that is not the case for Marek Michalak or many of the other investigators referenced in this study. I cannot make a recommendation for or against publication, but would like to state the following.

Thank you for your comments. I disagree with you that you are not competent to review this manuscript. The manuscript, for the first time, challenges the dogma that calreticulin was originally discovered as the HACBP in skeletal muscle SR. To support my hypothesis that the HACBP was an artefact and therefore the first publication on calreticulin was our 1984 paper, I have presented a retrospective study that I propose conclusively proves that the HACBP is an artifact. According to IntechOpen this is called a perspective review, which they define as–Perspective chapter (A perspective chapter offers a new viewpoint on existing problems, fundamental concepts, or prevalent notions on a specific topic. Perspective chapters can also propose or support new hypotheses, or discuss the implications of newly implemented innovations. Perspective chapters may focus on current advances and future directions on a topic and may include original data as well as personal opinion.)

Therefore, I am asking, respectfully,  that you review the evidence objectively and come to one of the following conclusions.

  1. Sufficient evidence is presented to challenge the dogma.

  1. Additional evidence is required before successfully challenging the dogma. If so, I would be grateful if you indicated what would be persuasive, as I have not presented all the evidence.

The second part of the review highlights some of our earlier studies --first to show CRT as an ER protein, first to sequence (N-terminal 15 residues) the protein--first to show conformational changes on metal binding--first to show the tissue distribution etc.--please note that none of these papers have been cited in the reviews on CRT. 

The third part of the review looks at our discovery that CRT is a plasminogen receptor and why that is significant.

I respectfully submit that it is important that the historical record of the discovery of CRT be factual.

Although not discussed in the manuscript, I would suggest that the term “high Affinity Calcium-Binding Protein” is inconsistent with the data presented in Ostwald and MacLennan (1974). The authors concluded that the HACBP bound one mol of Ca2+ with a Kd of between 2-4 uM and multiple Ca2+ with with lower affinity. This in not possible for several reasons and I state this from the vantage point of experience as I have performed Ca2+ binding studies on several proteins. First, the methods does not refer to any attempts to control for contaminating Ca2+ in the water and reagents. This is why we typically use ultra pure reagents, plasticware and not glassware  and chelex-treated water for these measurements. We have found that the contaminating levels of Ca2+ in these reagents will be at least 10-20 uM without mitigation. Therefore, measuring a 2-4 uM Kd under these experimental conditions is not possible. Second, the HACBP was not dialyzed against EDTA folowed by extenive dialysis against chelex-treated water. Therefore it is unclear what metal were already bound to the proteins. Finally, the Scatchard plot (Figure 5) shows a single slope–that means only one class of Ca2+-binding sites-not two as claimed by the authors. It is also interesting that Figure 4, the Ca2+ titration curve shows less Ca2+ binding at 5 uM than 10 uM–that speaks to the unreliability of these measurements. That figure also has no 0 point ( no added Ca2+) so it is unclear how much of the lower measurement are due to background.

I hope you will agree to evaluate the evidence that we have presented with an open mind. If you wish to challenge our hypothesis,, I would like to know what additional papers, other than the 1980 Michalak papers, support the existence of the HACBP. What I found most interesting is that I could not find any papers from labs that work on skeletal muscle SR proteins that have characterized CRT/HACBP in their preparations or have actually studied CRT in skeletal muscle SR. I know that Marek's usual response is that CRT is a minor SR protein--what has been published is that from 800g, one obtains 1mg of CRT after four chromatography columns (1974 paper) --that compares with 6 mg/800 g of calsequestrin which is purified with three columns.--definitely not a minor protein.

I apologize if my response is somewhat harsh--It only reflects my frustration and is not meant to be disrespectful.

  1.   Given that calreticulin is only marginally expressed in adult muscle, it does seem possible that Dr. Michalak came to the right conclusions for all of the wrong reasons, as it is unlikely that they would have been able to find such a strong calreticulin band in rabbit skeletal muscle.

Our perspective study cannot say why Michalak felt that he had sequenced the HACBP–just that many studies from labs that specialize in the study of SR proteins have been unsuccessful in identifying the HACBP/CRT in skeletal muscle.   If the HACBP actually exists then surely someone other than Michalak should have reported it. I actually found a reference to the HACBP in a review by Campbell who was a lab mate of Michalak–he doesn’t actually show it in a gel but refers to it as a 56 kDa protein that doesn’t stain with Stains all (Table II in his chapter). Therefore Michalak’s conclusion that CRT was initially discovered as the HACBP in skeletal muscle SR is unsubstantiated by any other laboratory. He actually sequenced calregulin. Therefore, what he got right I propose is that he sequenced calregulin. I know that he is aware of this because as described in his paper his sequence matched what we published for calregulin–he called me at that time requesting a sample of calregulin and our antibody. As he shows in his paper the HACBP antibody did not react against calregulin.

  1.   Calreticulin is an ER protein. I do not find the claims that it is a secreted protein in skeletal muscle only to be credible. Moreover, to the best of my knowledge, the claim that KDEL cannot be read in the sarcoplasmic reticulum to be false; KDEL receptors are neither in the ER or the SR. This does not mean that one never finds calreticulin outside of cells. There are many cellular components found in the extracellular milieu.

Agreed–we have removed the reference to KDEL receptors. I did not develop this suggestion very clearly–I meant to indicate that transport from the golgi retrograde to the ER has been well established, but to the best of my knowledge, golgi to SR has not been observed.

Round 2

Reviewer 2 Report

Comments and Suggestions for Authors

I've given this quite a bit of thought.

Whether or not CRT is a plasminogen receptor is a distinct topic from what seems to me to be the primary issue here: who identified CRT and based on what. As you have conceded my point that CRT is an ER/SR protein, a discussion on its potential extracellular roles seem to me to be a battle for another time. Preferably with someone else as the reviewer.

However, in reading your article, the idea that CRT could have been identified from skeletal muscle using ANY approach even with modern technology seems a bit unlikely. On that basis, I could be inclined to recommend publication of an article that does NOT reference other topics or directly state impropriety on the part of Dr. Michalak, as getting it right is definitely an important consideration...and they actually did get it right. For reasons not 100% clear.

Finally, if this article is to be published, I would like to write a commentary about it. My commentary will not be confused with an endorsement...but I won't support this publication in secret (ultimately, there are no secrets) or without commenting on why I would do such a thing.

Author Response

I've given this quite a bit of thought.

Thank you for your kind consideration.

Whether or not CRT is a plasminogen receptor is a distinct topic from what seems to me to be the primary issue here: who identified CRT and based on what. As you have conceded my point that CRT is an ER/SR protein, a discussion on its potential extracellular roles seem to me to be a battle for another time. Preferably with someone else as the reviewer.

Accordingly, we have removed the section on CRT as a plasminogen receptor and adjusted the graphical abstract accordingly.

However, in reading your article, the idea that CRT could have been identified from skeletal muscle using ANY approach even with modern technology seems a bit unlikely. On that basis, I could be inclined to recommend publication of an article that does NOT reference other topics or directly state impropriety on the part of Dr. Michalak, as getting it right is definitely an important consideration...and they actually did get it right. For reasons not 100% clear.

We are horrified by the suggestion from the reviewer that our review article implicated Dr. Michalak in any impropriety. I can assure the reviewer that this was not our intention. Scientific dogma often is often discarded in the light of subsequent discoveries. We simply disagree with Dr. Michalak’s assertion that the HACBP is CRT, period. The authors respectfully submit that progress in Science is built upon healthy debate of issues. Accordingly, we have added to the review the comment that “the authors distance themselves from any suggestion of impropriety by Dr. Michalak. This article is a perspective review which is defined as a new viewpoint on existing problems, fundamental concepts, or prevalent notions on a specific topic. Perspective chapters can also propose or support new hypotheses, or discuss the implications of newly implemented innovations. In this manuscript, we propose that the high-affinity Ca2+-binding protein was misidentified as a mixture of several proteins and does not exist as an actual protein. Therefore, the first report of CRT was that of calregulin, a Ca2+-binding protein of liver ER.

Finally, if this article is to be published, I would like to write a commentary about it. My commentary will not be confused with an endorsement...but I won't support this publication in secret (ultimately, there are no secrets) or without commenting on why I would do such a thing.

I look forward to reading your commentary. Thank you for taking the time to write it.